# Transcriptomic Analysis of the Anticancer Effects of Annatto Tocotrienol, Delta-Tocotrienol and Gamma-Tocotrienol on Chondrosarcoma Cells

**DOI:** 10.3390/nu14204277

**Published:** 2022-10-13

**Authors:** Kok-Lun Pang, Lian-Chee Foong, Norzana Abd Ghafar, Ima Nirwana Soelaiman, Jia Xian Law, Lek Mun Leong, Kok-Yong Chin

**Affiliations:** 1Department of Pharmacology, Faculty of Medicine, Universiti Kebangsaan Malaysia, Jalan Yaacob Latif, Bandar Tun Razak, Cheras 56000, Malaysia; 2State Key Laboratory of Oncogenes and Related Genes, Renji-Med X Clinical Stem Cell Research Center, Department of Urology, Ren Ji Hospital, School of Medicine, Shanghai Jiao Tong University, Shanghai 200127, China; 3Department of Anatomy, Faculty of Medicine, Universiti Kebangsaan Malaysia, Jalan Yaacob Latif, Bandar Tun Razak, Cheras 56000, Malaysia; 4Centre for Tissue Engineering and Regenerative Medicine, Universiti Kebangsaan Malaysia Medical Centre (UKMMC), Jalan Yaacob Latif, Bandar Tun Razak, Cheras 56000, Malaysia; 5Prima Nexus Sdn. Bhd., Suite 8-1 & 8-2, Level 8, Menara CIMB, Jalan Stesen Sentral 2, Kuala Lumpur 50470, Malaysia; 6Department of Biomedical Science, Faculty of Science, Lincoln University College, Wisma Lincoln, No. 12-18, Jalan SS 6/12, Petaling Jaya 47301, Malaysia

**Keywords:** vitamin E, apoptosis, paraptosis, endoplasmic reticulum, cell cycle arrest, anti-proliferation

## Abstract

Previous studies have demonstrated the anticancer activities of tocotrienol on several types of cancer, but its effects on chondrosarcoma have never been investigated. Therefore, this study aims to determine the anticancer properties of annatto tocotrienol (AnTT), γ-tocotrienol (γ-T3) and δ-tocotrienol (δ-T3) on human chondrosarcoma SW1353 cells. Firstly, the MTT assay was performed to determine the half-maximal inhibitory concentration (IC_50_) of tocotrienol on SW1353 cells after 24 h treatment. The mode of cell death, cell cycle analysis and microscopic observation of tocotrienol-treated SW1353 cells were then conducted according to the respective IC_50_ values. Subsequently, RNAs were isolated from tocotrienol-treated cells and subjected to RNA sequencing and transcriptomic analysis. Differentially expressed genes were identified and then verified with a quantitative PCR. The current study demonstrated that AnTT, γ-T3 and δ-T3 induced G1 arrest on SW1353 cells in the early phase of treatment (24 h) which progressed to apoptosis upon 48 h of treatment. Furthermore, tocotrienol-treated SW1353 cells also demonstrated large cytoplasmic vacuolation. The subsequent transcriptomic analysis revealed upregulated signalling pathways in endoplasmic reticulum stress, unfolded protein response, autophagy and transcription upon tocotrienol treatment. In addition, several cell proliferation and cancer-related pathways, such as Hippo signalling pathway and Wnt signalling pathway were also significantly downregulated upon treatment. In conclusion, AnTT, γ-T3 and δ-T3 possess promising anticancer properties against chondrosarcoma cells and further study is required to confirm their effectiveness as adjuvant therapy for chondrosarcoma.

## 1. Introduction

Vitamin E or tocochromonal is a fat-soluble vitamin found in several foods, including nuts and vegetable oils. There are two different classes of tocochromanols, namely tocopherol and tocotrienol. Tocopherol has a saturated aliphatic phytyl tail and tocotrienol has a tri-unsaturated phytyl tail. They are further divided into α, β, γ and δ-isomers, depending on the number and degree of hydroxyl and methyl groups on the chromanol ring [1]. Vitamin E isomers are naturally present in a mixture and vary among plant sources. For instance, palm oil is one of the most abundant natural sources of vitamin E (with 600–1000 mg/kg) which consists of mainly 37–43% γ-tocotrienol, 24% α-tocotrienol and 21% α-tocopherol [2,3,4]. On the other hand, rice bran oil contains lesser vitamin E (around 300 mg/kg) with around 25% tocopherols, 17% tocotrienols and a negligible amount of γ-tocotrienol (γ-T3) [5]. Vitamin E extracted from annatto beans is unique in composition with the absence of tocopherol but 10–16% γ-T3 and 84–90% of δ-tocotrienol (δ-T3) [6,7]. Plant oil-derived tocotrienol or tocotrienol-rich fraction (TRF) is a more viable candidate for commercialization than their pure isomers which require complicated and expensive isolation procedures [5].

Vitamin E, especially tocotrienol or TRF, is known for its antioxidant and anti-inflammatory properties. Besides, it possesses broad biological activities in preventing or treating several non-communicable diseases, including anti-diabetic, cholesterol-lowering, anti-lipidemic, anti-osteoporotic and anticancer activities [8]. Tocotrienol and TRF are potent anticancer agents, wherein they induce anti-proliferation, growth arrest, apoptosis and/or autophagy in several cancerous cell lines from the origin of the mammary gland [9,10], colon [11,12,13], lung [14], liver [15], prostate [16], melanoma [17,18,19], stomach [20] and leukemic cells [21]. Similar promising results were also reported in the animal cancer models, xenografts [22,23,24] and clinical trials [25]. Mechanistically, tocotrienols, especially γ-T3 and δ-T3, exert multiple modes of action to execute cancerous cells, i.e., by inducing endoplasmic reticulum (ER) stress [19,26,27], cell cycle arrest [28], mitogen-activated protein kinase (MAPK) activation [28] and the suppression of Wnt signalling [29,30], phosphoinositide 3-kinase (PI3k)/Akt and/or nuclear factor-κB pathways [31]. Furthermore, tocotrienol demonstrates minimal cytotoxic effects on several non-cancerous cell lines [32,33]. It is generally safe with no observed adverse effects at the dose of 120–130 mg/kg/day for rats [34]. In addition, tocotrienol is well tolerated at the dose of ≤400 mg/day in humans without any genotoxic, mutagenic, carcinogenic or teratogenic effects [35,36,37,38]. A recent study showed that 12-week supplementation of annatto tocotrienol up to 830 mg/day did not cause any adverse event in postmenopausal women with osteopenia [39].

Chondrosarcoma is a type of malignant primary bone tumour that involves cartilage cells. It is the second most common type of primary bone cancer and relatively rare compared with other malignancies [40]. According to Netherland Cancer Registry, the incidence of chondrosarcoma has increased from 2.88/million in 1989–1996 to 8.78 in 2005–2013 [41]. Chondrosarcoma primarily affects the cartilage cells of the proximal femur, pelvis, upper arm bone, shoulders or ribs. Like many other cancers, the aetiology of chondrosarcoma is unknown. Chondrosarcoma commonly occurs in patients above 40 years old, and the risk increases with age [40,42]. Several bone conditions, such as enchondroma, osteochondroma and Maffucci syndrome, may predispose to the formation of chondrosarcoma [43].

Chondrosarcoma is usually diagnosed accidentally due to its rare incidence and non-specific signs and symptoms. The early stage of chondrosarcoma can be asymptomatic, and patients in the intermediate or advanced stage may suffer from deep and progressive joint pain, especially at night with joint dysfunction or bone fracture [40,44]. To date, no blood test or established biomarker is clinically used in diagnosis [45]. Chondrosarcoma lesions can be detected by several imaging techniques, including radiograph, computerized tomography or magnetic resonance imaging scanning at the affected sites [40,44]. A definitive diagnosis is made via biopsy with the histological detection of invasive malignant cells with hypercellular stroma. Most of the chondrosarcoma cases (85–90%) are of low or intermediate grade (classified as central conventional chondrosarcoma) with limited metastatic ability [40,44]. The prognosis of chondrosarcoma depends on its grade or stage, location and age. Dedifferentiated chondrosarcoma is highly lethal where most of the patients died within the first 2 years with only a 10–13% survival rate at 5 years [46]. Early detection and treatment of low-grade chondrosarcoma can achieve a better prognosis with a 75–80% survival rate at 5 years [42,47,48,49,50]. Treatment of chondrosarcoma is generally by wide surgical resection or intra-lesional curettage [51]. Nevertheless, chondrosarcoma, including the low grade, is resistant to chemotherapy and radiotherapy [40,44,45]. Radiotherapy or chemotherapy are used as adjuvant or palliative therapy in high-grade or dedifferentiated chondrosarcoma. Therefore, there is a need to develop a better and more effective strategy for managing chondrosarcoma, especially in the advanced stage.

To the best of our knowledge, the anticancer effects of tocotrienol on chondrosarcoma are yet to be determined. Since new anticancer candidates in chondrosarcoma chemotherapy are needed, this study aims to determine the anticancer effects of γ-T3 and δ-T3 on human chondrosarcoma SW1353 cells. This study also assessed whether annatto-derived tocotrienol (AnTT) containing γ-T3 and δ-T3 could exert similar chemotherapeutic actions as pure tocotrienol isomers. Cytotoxicity, mode of cell death and cell cycle analysis were conducted to determine the anticancer properties of AnTT, γ-T3 and δ-T3 on SW1353 cells. Subsequently, Gene Ontology (GO) and Kyoto Encyclopaedia of Genes and Genomes (KEGG) pathway enrichment analyses were performed to analyse the RNA-sequencing data and the underlying pathways. We hypothesised that AnTT, γ-T3 and δ-T3 could induce anti-proliferation and apoptosis events on SW1353 cells by activating several cell death pathways. The findings of this study will provide insights on the potential application of tocotrienol as a new candidate in adjuvant therapy of chondrosarcoma.

## 2. Materials and Methods

### 2.1. Cell Lines

Human chondrosarcoma SW1353 cells were purchased from American Type Culture Collection (catalogue no. HTB 94) and cultured with high glucose Dulbecco’s modified Eagle’s medium (DMEM; Nacalai Tesque, Japan; catalogue no. 08458-16) that contains 4 mM L-glutamine, 1 mM sodium pyruvate, 10% FBS (Thermo Fisher Scientific, Waltham, MA, USA; catalogue no. 10270-106) and 1% antibiotic-antimycotic solution (Thermo Fisher Scientific, Waltham, MA, USA; catalogue no. 15240-062) at 37 °C in an incubator with humidified air and 5% carbon dioxide. These cells were passaged every 2 days to maintain the cells in logarithmic growth. The cells in early passages (3rd–10th passages) were used in the experiment.

### 2.2. Materials

All the chemicals used in this study were purchased from Sigma-Aldrich unless stated otherwise. AnTT, pure γ-T3 and δ-T3 were provided by American River Nutrition (Hadley, USA) where AnTT contains 84% δ-T3 and 16% γ-T3 (Lot Number: #18FA-1270, purity 70%). Stock solutions of AnTT, γ-T3 and δ-T3 were prepared according to our previous studies with slight modifications in concentration [52,53]. Briefly, AnTT, δ-T3 and γ-T3 stock solutions were firstly dissolved in absolute ethanol to a concentration of 0.1 g/mL. All stock solutions were then aliquoted and kept at −80 °C until use. The day before treatment, 45 µL of tocotrienol stock solution was mixed with 60 µL sterile foetal bovine serum (FBS) and incubated at 37 °C for 24 h. After that, 105 µL of absolute ethanol was added to the mixture to mix well and followed by 90 µL of complete cell culture media. The same procedure was employed for the vehicle control (VC) group by using 45 µL of absolute ethanol. The entire process of tocotrienol stock preparation, storage, incubation and treatment were protected from direct light exposure.

### 2.3. Cytotoxicity Assay

The cytotoxicity of tocotrienols was determined using the yellow tetrazolium salt 3-(4,5-dimethylthiazol-2-yl)-2,5-diphenyltetrazolium bromide (MTT) according to previous studies with slight modification [6]. Briefly, 100 µL of human chondrosarcoma SW1353 cells with a density of 5 × 10^4^ cells/mL were seeded in a 96-well plate with complete media for 24 h. On the next day, the media were removed and replaced with 100 µL of complete media with a series of concentrations of tocotrienol (10–50 µM) for another 24 h. At the end of the treatment, 20 µL of MTT solution (5 mg/mL) was added to each well and then further incubated for another 4 h inside a 37 °C incubator. The optical density (OD) of each well was measured at 570 nm by using a Multiskan GO microplate reader (Thermo Fisher Scientific, Vantaa, Finland). The viability of the treated cells was calculated by dividing the OD of the treated group with the OD value of the VC. The values of 50% maximal inhibitory concentration (IC_50_) were determined manually from the graph and used for subsequent experiments.

The antiproliferative effects of tocotrienols on SW1353 cells were determined using a modified MTT assay with a low cell number but with a longer incubation time. Briefly, 100 µL of SW1353 cells (1 × 10^4^ cells/mL) were seeded in a 96-well plate for 24 h. Subsequently, the media were replaced with 100 µL of fresh media with a series of concentrations of tocotrienols (0.625–10 µM) for 72 h. Standard MTT assay was conducted at the end of the treatment. The viability of cells was calculated as a percentage relative to the VC. The IC_50_ values for the growth inhibition were determined manually from the graph to compare the potency of antiproliferation among tocotrienols.

### 2.4. Annexin V-FITC/PI Dual-Labelling Assay

The mode of cell death, by either apoptosis or necrosis, was determined according to the plasma membrane surface externalization of phosphatidylserine and its integrity. The staining procedure was performed according to our previously reported workflow [54]. Briefly, 3 mL of SW1353 cells (5 × 10^4^ cells/mL) were seeded in a 6-well plate for 24 h, followed by treatment with IC_50_ values of tocotrienols for another 24 and 48 h. Before collecting the cells, the images of cells were captured at 200× magnifications using an Olympus CKX31 inverted microscope with an X-CAM alpha camera and DigiAcquis 2.0 acquisition software (Matrix Optics, Petaling Jaya, Malaysia) for morphological comparison [54]. Then, all the treated cells including the floated cells were collected and washed twice with ice-cold phosphate-buffered saline (PBS) at 200× *g* for 5 min. The cells were resuspended in 350 µL of annexin V staining buffer (BD Bioscience, San Jose, CA, USA) and stained with 5 µL of annexin V conjugated with fluorescein isothiocyanate (annexin V-FITC) (BD Bioscience, San Jose, CA, USA) on ice and dark condition for 15 min. A 10 µL of 50 µg/mL propidium iodide (PI) solution was then added to the cells and incubated for another 5 min. After the dual staining, another 150 µL of annexin V staining buffer was added to the cells and transferred to flow tubes. A total of 10,000 stained cells were recorded and analysed by using BD FACSVerse flow cytometer (BD Bioscience, San Jose, CA, USA). The cell population with negative staining for both markers represents viable cells population, positive staining for annexin V-FITC represents an early apoptotic population, positive staining for propidium iodide represents the necrotic population whereas positive staining for both markers represents a late apoptotic population. Both early and late apoptotic populations were collectively reported as the apoptotic population in this study.

### 2.5. DNA Content Assay

The DNA content or cell cycle analysis was conducted as described [54,55]. Briefly, 3 mL of SW1353 cells (5 × 10^4^ cells/mL) were seeded in a 6-well plate for 24 h, followed by the IC_50_ of tocotrienols treatment for 12 to 24 h. A serum-free condition was used as cell cycle arrest control. Then, all the treated cells including the floated cells were collected, trypsinized and washed twice with ice-cold PBS at 200× *g* for 5 min. The cells were then counted, and 1 × 10^5^ cells were fixed with 1 mL of 70% ethanol, added drop by drop while vortexing and kept for overnights. Then, the fixed cells were washed again with ice-cold PBS at 400× *g* for 10 min. Subsequently, the cells were stained with 500 µL of PI staining solution (BD Bioscience, Franklin Lakes, NJ, USA) for 15 min at room temperature. At least 12,000 stained cells were recorded and analysed by using BD FACSVerse flow cytometer (BD Bioscience, Franklin Lakes, NJ, USA). The cell cycle distribution was further analysed by using ModFit LT^TM^ software (Verity Software House, Topsham, ME, USA) via Gaussian curves modelling estimation [56,57].

### 2.6. RNA Isolation

The RNA sample preparation was conducted using the RNeasy Mini Kit (Qiagen, Hilden, Germany) according to the manufacturer’s instructions. Briefly, 6.5 mL of SW1353 cells with a density of 5 × 10^4^ cells/mL were seeded in a 60 mm culture dish for 24 h. Subsequently, the SW1353 cells were challenged with tocotrienols with their respective IC_50_ values for 24 h. Then, the media with the floated cells were collected and combined with the trypsinised cells. The entire mixture was centrifuged and washed twice with ice-cold PBS at 200× *g* for 5 min. The cell pellets were then mixed with 350 µL of Buffer RLT for lysis. After that, the lysate was mixed with 350 µL of 70% ethanol by pipetting and then vortex for 1 min. The entire content was transferred to a RNeasy Mini spin column with a collection tube and then centrifuged at 8000× *g* for 15 s. The flow-through collected in the collection tube was discarded. The procedure was continued with the addition of 700 µL of Buffer RW1 to the spin column and then the spin column was centrifuged again at 8000× *g* for 15 s. The flow-through was discarded and then 500 µL of Buffer RPE was added to the spin column and then the spin column was centrifuged again at 8000× *g* for 15 s. The flow-through was discarded again. Another 500 µL of Buffer RPE was added to the spin column and the spin column was centrifuged at 10,000× *g* for 2 min. The collection tube with flow-through was discarded and replaced with a new tube. The spin column was centrifuged again at 10,000× *g* for 1 min to completely remove the trace Buffer RPE. The collection tube with flow-through was discarded again and replaced with a new 1.5 mL collection tube. A 50 µL of RNase-free water was added to the spin column and the spin column was centrifuged at 10,000× *g* for 1 min to collect the eluted RNA. The purity of RNA was determined according to the absorbance ratio of 260 nm to 280 nm (A260/280) using BioTek^TM^ PowerWave HT microplate spectrophotometer with a nanodrop adapter (BioTek, Winooski, VT, USA). RNA samples were aliquoted and kept at −80 °C until further analysis.

### 2.7. RNA Sequencing and Transcriptomic Analysis

The purity and integrity of RNA samples were evaluated using Agilent RNA 6000 Bioanalyzer with Agilent 2100 Expert software and nanodrop. Only the RNA samples with RNA Integrity Number (RIN) ≥7 and A260/280 ≥2 proceeded to subsequent RNA sequencing. Then, 100 ng of total RNA per sample was subjected to DNase treatment and converted to cDNA by 2-step reverse transcription. Fragmentation of cDNA was conducted, and a unique barcoded adaptor (8 bp) was added to each sample for subsequent multiplex sequencing. Human AnyDeplete technology was used during second strand selection to deplete the ribosomal RNA (rRNA) and improve the sequencing efficiency. The strand-specific rRNA-depleted cDNA libraries were then amplified by 14 cycles of PCR. The final quality of cDNA libraries (size distribution and concentration) was confirmed by Agilent Technologies Bioanalyser DNA High Sensitivity assay where a human reference RNA was used as an internal control. The RNA sequencing was performed using the NuGEN Universal RNA-Seq system for paired-end sequencing on the Illumina platform. Raw RNA sequencing data were generated in a FASTQ file format.

### 2.8. Differential Expression Analysis and Enrichment Analysis

FASTQ files were pre-processed using Trim Galore software (version 0.4.5) with the default setting to remove reads with adapter, errors or low quality. The trimmed reads data were then aligned to the GENCODE human reference genome sequences (GRCh38.p13) using STAR (version 2.6.0a) and exported as BAM files. Subsequently, raw gene counts were obtained using the Bioconductor RSubread package (version 2.10.4) [58]. Genes with less than 12 reads were excluded from further analysis. Principal component analysis (PCA) was applied to understand key properties of the datasets including the variations among the groups of sample replicates. Differentially expressed genes (DEGs) between VC and tocotrienol treatments or among respective tocotrienol treatments were identified with the Bioconductor DESeq2 package (version 1.36.0) [59] using the default parameters. Transcripts with adjusted *p*-value < 0.05 and absolute log_2_ transformed fold change value (|log_2_FC|) > 1 were defined as DEGs. The DEGs were subsequently subjected to gene set over-representation analysis associated with Gene Ontology (GO) and Kyoto Encyclopaedia of Genes and Genomes (KEGG) database using the clusterProfiler R package (version 4.4.4). Data visualization was performed using the enrichplot (version 1.16.1; [60]), pheatmap (version 1.0.12; [61]), and ggplot2 (version 3.3.6; [62]) R packages. Genes associated with the top enriched KEGG pathways were illustrated using the R function cnetplot of the clusterProfiler package.

### 2.9. Quantitative PCR (qPCR) Validation

The RNA-sequencing data were then compared and validated by using qPCR, based on 5 selected genes with 4 genes (*ARMCX3*, *HMGCS1*, *ADAMTSL1* and *KLHDC7B*) randomly selected from the top 20 upregulated and downregulated DEGs and *GAPDH* gene as a housekeeping gene. Briefly, 1 μg of total RNA from each sample was converted to cDNA using Agilent AffinityScript QPCR cDNA Synthesis kit. The cDNA samples were then quantified, and 10 ng of cDNA was used in qPCR assay using Agilent Brilliant III Ultra-Fast SYBR^®^ Green QPCR Master Mix according to the manufacturer’s protocol. The reverse and forward primers for the 5 targeted genes were designed using the Primer3 tool with a primer melting temperature of 60 °C. The full sequences of the primers are listed in Appendix A. The reaction mixtures were then cycled in an Agilent MX3005P real-time PCR system with a denaturation step at 95 °C for 3 min, followed by 40 cycles of annealing/extension with 95 °C for 5 s and 60 °C for another 20 s, according to the manufacturer’s protocol. The fluorescence signals were detected at each cycle during the 60 °C annealing/extension step. The threshold cycle (CT) values were determined from the respective amplification curves using Agilent MXPro QPCR software. The fold change was determined by using the 2^−∆∆CT^ method.

### 2.10. Statistical Analysis

Statistical analysis was conducted using the SPSS software for Windows, version 25 (IBM, Armonk, NY, USA). The normality of data was tested by the Shapiro–Wilk test. Mean differences between multiple groups were analysed using a one-way analysis of variance with Tukey or Dunnett’s T3 post hoc analysis. The concordance of RNA-sequencing data and qPCR was determined with correlation analysis by calculating the value of coefficient of correlation (R^2^). A *p*-value < 0.05 is considered statistically significant.

## 3. Results

### 3.1. Cytotoxicity of AnTT, γ-T3 and δ-T3 on Human Chondrosarcoma SW1353 Cells

Cytotoxicity of AnTT, γ-T3 and δ-T3 on SW1353 cells over 24 h treatment was evaluated using the MTT assay (Figure 1). The current findings revealed that AnTT, γ-T3 and δ-T3 were cytotoxic to SW1353 cells in a concentration-dependent manner with IC_50_ values of 28.5, 38.5 and 19.5 μg/mL, respectively. Among the treatment, δ-T3 is the most potent cytotoxic agent in reducing the viability of SW1353 cells, followed by AnTT, and lastly γ-T3. The respective IC_50_ values were used in the subsequent experiment.

### 3.2. The Mode of SW1353 Cell Death upon AnTT, γ-T3 and δ-T3 Treatment

The mode of SW1353 cell death was assessed upon the AnTT, γ-T3 and δ-T3 treatment with their respective IC_50_ values for 24 and 48 h (Figure 2). The 24 h tocotrienol treatments did not induce significant cell death on SW1353 cells at their respective IC_50_ concentration. Significant cell death, mainly through apoptosis, was detected when the treatment time was prolonged to 48 h. AnTT, γ-T3 and δ-T3 killed around 50%, 70% and 25% of the cell population, respectively, as indicated by plasma membrane phosphatidylserine externalization. All tocotrienol treatments caused plasma membrane rupture or necrosis in <2% of the cell population.

### 3.3. Morphological Changes of SW1353 Cells upon 24 h and 48 h AnTT, γ-T3 and δ-T3 Treatment

The morphology of SW1353 cells after 24 h and 48 h of tocotrienol treatments was also observed (Figure 3). SW1353 cells treated with VC assumed typical spindle fibroblast-like morphology. Consistent with cytotoxicity and mode of cell death findings, all tocotrienol treatments caused the SW1353 cells to shift to elongated or squamous-like morphology with occasionally cell shrinkage. In addition, the number of attached cells upon tocotrienol treatments was also reduced. On the other hand, 48 h of tocotrienol treatments also caused a similar morphological alteration with reduced cell density and abnormal cellular shape. In addition, massive vacuoles were observed in SW1353 cells after 24 h and 48 h of tocotrienol treatments. γ-T3 treatment induced more severe cell death with prominent cell shrinkage and cell debris after 48 h of tocotrienol treatments. On the other hand, 48 h δ-T3-treated SW1353 cells were squamous or elongated in shape with massive vacuolation but with lesser cell shrinkage.

### 3.4. Antiproliferative and Growth Arrest Action of AnTT, γ-T3 and δ-T3 on SW1353 Cells

Tocotrienol induced growth arrest on SW1353 cells in the early phase and cell death in the later stage. Therefore, a modified MTT assay was performed to evaluate the effects of tocotrienol treatment on the proliferation of SW1353 cells. Figure 4 showed the antiproliferative properties of AnTT, γ-T3 and δ-T3 on SW1353 cells for 72 h treatment with IC_50_ of 8.6, 9.4 and 8.1 μg/mL. Parallel with the 24 h viability data, δ-T3 demonstrated the most potent antiproliferation activity on SW1353 cells. The subsequent cell cycle analysis also revealed that AnTT, γ-T3 and δ-T3 induce G1 arrest on SW1353 cells significantly upon 24 h treatment (Figure 5 and Figure 6). Serum deprivation was used in this experiment as a validation control by inducing G1 arrest (*p* < 0.001).

### 3.5. RNA Isolation and Quality

The RNA samples were isolated from SW1353 cells with vehicle (VC), AnTT, γ-T3 and δ-T3 treatment for 24 h. The quality and quantity of isolated RNAs were checked before cDNA synthesis and the results were summarised in Appendix A. Generally, all the RNA samples are of good quality with RIN of 8.1–9.8 and A260/280 of 2.05–2.07.

### 3.6. Data Generation and Quality of RNA Sequencing Reads

RNA sequencing by Illumina HiSeqX was conducted on the RNA samples from SW1353 cells upon treatment. The total reads, trimmed reads, low-quality reads with Phred score of less than 20, clean reads and mapped reads for all samples were summarized in Appendix A. In summary, there were 12,757,624 to 16,239,630 clean reads (≥99.25%) for all RNA samples, indicating good quality of clean reads. Additionally, the aligned mapping rate for all samples was generally >80% which is qualified for the subsequent DEGs identification and transcriptomic analysis [63].

### 3.7. DEGs Identification upon Tocotrienol Treatment

Using 28,879 genes with non-zero total read count, principal component analysis (PCA) revealed a clear distinction of the gene expression between cells treated with three different tocotrienol treatments and VC, and consistency between replicates within each treatment (Figure 7A). The scree plot illustrates the dominance of PC1 to explain the variation of the dataset (Figure 7B). Together, the first two PCs describe 98% of the variations in the dataset Read counts were modelled using a negative binomial generalized linear model (GLM) in DESeq2 to compare the effects of each treatment with vehicle control (VC) and between treatment groups. We identified the DEGs with the cut-off criteria of |log_2_FC| >1 and adjusted *p*-value < 0.05. A total of 2508 DEGs were identified between the AnTT and VC groups, in which 1123 DEGs were upregulated and 1385 DEGs were downregulated in AnTT-treated SW1353 cells. In addition, 3577 DEGs were found between γ-T3 and VC groups, whereby 1613 DEGs were upregulated and 1964 DEGs were downregulated in γ-T3-treated cells. A total of 3929 DEGs were also detected between the δ-T3 and VC groups, whereby 1670 DEGs were upregulated and 2259 DEGs were downregulated in δ-T3-treated cells. The volcano plots and heatmap reflect the gene expression profiles of the DEGs (Figure 7C–F). Interestingly, as demonstrated in the Venn diagrams for these three group comparisons (Figure 8), a large amount of DEGs overlapped, suggesting that AnTT treatment as well as the pure tocotrienol treatments exhibited similar regulations on SW1353 cells. Besides that, no up- or down-regulated DEG was detected between tocotrienol treatments, except six upregulated DEGs (*ACKR3, ATOH8, CA12, LTBP3, PLXNA4, RGS4*) were spotted in AnTT-treated cells compared with δ-T3-treated cells. The complete list of up- and down-regulated DEGs was summarized in Appendix A.

### 3.8. GO Enrichment Analysis

To further understand the functional roles of the DEGs, GO enrichment analysis was performed using clusterProfiler. Enrichment analysis showed a total of 2015 GO-terms were significantly enriched with 59 and 1086 enriched terms in up- and down-regulated AnTT-vs-VC DEGs; 132 and 1404 enriched terms in up- and down-regulated γ-T3-vs-VC DEGs; and 102 and 1748 enriched terms in up- and down-regulated δ-T3-vs-VC DEGs, respectively (Appendix A). As shown in Figure 9, the AnTT, γ-T3 and δ-T3 treatment resulted in similar enriched GO categories and subcategories of DEGs, except results in the molecular function (MF) subcategory. In the biological process (BP) and cellular component (CC) subcategories, these three treatments shared similar top enriched GO terms, which mainly related to the activities of endoplasmic reticulum- (ER) and Golgi-related activities, including response to ER stress, Golgi vesicle transport, cilium organization, ER unfolded protein response, cilium assembly, and ER to Golgi vesicle-mediated transport. Interesting, only upregulated DEGs in γ-T3 treatment showed significant GO molecular functions, namely SNAP receptor activity, catalytic activity, acting on RNA, chaperone binding, and SNARE binding. The downregulation of DEGs in all three treatments was similar in all top GO terms for all GO subcategories. In the BP subcategories, the downregulated DEGs mainly participated in actin filament-related activity, protein-DNA complex assembly, and ossification. These biological processes were coherent to the top enriched CC subcategories, which showed enrichment of DEGs in the actin cytoskeleton, cell–cell junction, focal adhesion, DNA packaging complex, and nucleosome. In the MF subcategories, the downregulated DEGs were found to be enriched in the protein heterodimerization activity, several complex bindings, including actin-, cadherin-, actin filament-, and beta-catenin binding, as well as structural constituents of the cytoskeleton, which is consistent with the results in the BP subcategories and our observation of the morphological changes upon treatment.

### 3.9. KEGG Pathway Enrichment Analysis

The top KEGG enriched pathways for respective tocotrienol treatments were summarized in Figure 10 and Appendix A. Similar to GO enrichment data, the enriched KEGG pathways were similar among the downregulated DEGs of AnTT, δ-T3 and γ-T3 treatments. DEGs associated with neutrophil extracellular trap formation, cell cycle, Hippo signalling pathway, Rap1 signalling pathway as well as some cancer-related pathways (e.g., bladder, breast, colorectal, gastric, non-small cell and small cell lung, pancreatic, prostate) were consistently downregulated in tocotrienol treatments compared to VC treatment. Besides, other downregulated DEGs were related to TGF-beta signalling pathway, p53 signalling pathway, Wnt signalling pathway, focal adhesion, and PI3k-Akt signalling pathway. On the other hand, the common enriched KEGG pathways for the upregulated DEGs between tocotrienol treatments and VC were identified as protein processing in ER, autophagy, mitophagy, protein export, and SNARE interactions in vesicular transport. Besides that, the upregulated DEGs each treatment has found uniquely enriched in several KEGG terms, which included but not limited to, pathways of neurodegeneration, amyotropic lateral sclerosis, amino sugar and nucleotide sugar metabolism, and collecting duct acid secretion. These critical pathways along with the related DEGs, as presented in Figure 11, are worth further investigations to discover the unique mechanism through which these treatments affect the cells.

### 3.10. qPCR Validation

The findings from RNA sequencing were further validated with qPCR. Four DEGs in the top 20 up- and downregulated lists were randomly selected with the *GAPDH* gene as a housekeeping gene. The KLHDC7B and ARMCX3 genes are the up-regulated DEGs while HMGCS1 and ADAMTSL1 genes are the downregulated DEGs. The FC values were calculated based on the 2^−∆∆CT^ approach. Table 1 summarised the FC values between qPCR and RNA sequencing data. Generally, all the RNA sequencing data for ARMCX3, HMGCS1 and KLHDC7B genes were coherent with qPCR data, except for ADAMTSL1. The FC values for ARMCX3, HMGCS1 and KLHDC7B genes were coherent with RNA sequencing data for tocotrienol treatments as compared to VC. The discrepancy of FC values for ADAMTSL1 is probably due to the presence of multiple transcript variants from alternative splicing [64]. The FC values for ARMCX3, HMGCS1, KLHDC7B and ADAMTSL1 genes from qPCR were between 0.5 and 2. These genes are not listed as DEGs under RNA sequencing because they are excluded in the first place with FC values of 0.5 to 2. Correlation analysis was conducted to determine the correlation of the FC values from qPCR and RNA sequencing. The data points without absolute FC values (RNA sequencing; labelled as N.A) were excluded from this correlation analysis. Figure 12 demonstrated a strong agreement between qPCR and RNA sequencing data with R^2^ value of 0.9827 or 98.27%. This indicated the good reliability of the RNA sequencing result.

## 4. Discussion

The current study demonstrated that AnTT, δ-T3 and γ-T3 possess anticancer properties by suppressing the viabilities of human chondrosarcoma SW1353 cells with IC_50_ values of 28.5, 38.5 and 19.5 μg/mL, respectively. To our best knowledge, this is the first demonstration of anticancer properties of AnTT, purified δ-T3 and γ-T3 on human chondrosarcoma. The potencies of tocotrienols on chondrosarcoma cells are comparable with most of the other in vitro cancer cell lines with a similar IC_50_ range of 15 to 25 μg/mL at 24 h treatment (Table 2).

To our surprise, in this study, all 24 h tocotrienol treatments did not induce significant cell death, regardless of apoptosis or necrosis despite IC_50_ treatment. The discrepancy between the MTT assay and annexin V-FITC/PI labelling assay is potentially due to the different end-point measurements. The viability results from the MTT assay are greatly dependent on the number of functional mitochondria in viable cells that convert the yellow MTT reagent to dark purple formazan. On the other hand, the annexin V-FITC/PI labelling assay determines the viability of cells based on the membrane integrity and the presence of phosphatidylserine (early apoptotic marker) [54]. In this scenario, the 24 h-tocotrienol treatments may induce growth arrest on SW1353 cells by reducing the number of viable cells without triggering the cell death event. Further investigation on the cell cycle analysis confirmed this speculation, whereby AnTT, γ-T3 and δ-T3 significantly induced G1 arrest upon 24 h treatment (Figure 5 and Figure 6). Similarly, a modified MTT assay also revealed the antiproliferative activities of AnTT, γ-T3 and δ-T3 on SW1353 cells with IC_50_ of 8.6, 9.4 and 8.1 μg/mL, respectively (Figure 4). Furthermore, prolonging the tocotrienol treatment to 48h did induce prominent cell death events on SW1353 cells, primarily through apoptosis (Figure 2B). Collectively, AnTT, γ-T3 and δ-T3 induced early G1 arrest on SW1353 cells which progressed to apoptosis with prolonged treatment time.

Previous studies had reported similar findings of tocotrienol in inducing early growth arrest, followed by cell death in several cancer cells [65,66,73,75]. AnTT and δ-T3 did not induce significant apoptosis events on human breast cancer SKBR3 with their respective IC_50_ values [65]. In addition, AnTT also upregulated the cell cycle arrest proteins such as p53, p21^WAF1^ and p27^kip1^ in human breast cancer SKBR3 cells upon 24 h treatment [65]. Similarly, 24 h of AnTT treatment also induced G1 arrest on human prostate cancer LNCaP cells and only caused significant apoptosis events upon 48 h treatment [66]. δ-T3 also induced G1 arrest on human melanoma A2058 cells with the reduced expression of cyclin-dependent kinase 4 [73]. β-tocotrienol also induced G1 growth arrest on human breast cancer MCF-7 cells where an apoptotic event was only triggered upon a higher concentration (higher than IC_50_) or prolonged treatment time [75].

Several comparative studies had shown the differences in the potency of anticancer activities among tocotrienol isomers in the order of δ > γ > β > α [1,76]. Moreover, γ-T3 and δ-T3 could achieve better cellular transport and accumulation compared to other tocotrienol isomers or tocopherols [1,66] and uniquely accumulated in the tumour tissues [77,78]. In this study, δ-T3 is the most potent tocotrienol with the lowest IC_50_ values in the MTT assay (Figure 1) and antiproliferation assay (Figure 4), followed by AnTT and γ-T3. Although the potency of AnTT is between the γ-T3 and δ-T3, the precise interaction between isomers like addition or synergism is unknown. The interaction between tocotrienol isomers is not reported so far. The combination of AnTT with γ-tocopherol significantly increased its potential in suppressing the viability of human prostate cancer LNCaP cells [66]. In addition, a potential antagonism interaction from α-tocopherol against tocotrienol was suggested as well [79,80]. Regardless of the underlying interaction between γ-T3 and δ-T3, current findings showed the promising anticancer effect of AnTT on chondrosarcoma. In addition, AnTT is a better anticancer candidate because its production cost is cheaper than individual tocotrienol isomers.

Paraptosis is a mode of unique programmed cell death independent of caspases activation and lack of typical apoptotic features [81]. Additionally, cells undergoing paraptosis have an intact cell membrane, phosphatidylserine markers, extensive cytoplasmic vacuolation and swelled ER and mitochondria [81]. In this study, the microscopic observation revealed an interesting finding, whereby AnTT, γ-T3 and δ-T3 induced massive vacuole formation on SW1353 cells which may indicate activation of paraptosis or paraptosis-like cell death. γ-T3 [30,82] and δ-T3 [83,84] were previously reported to induce paraptosis in several cancer cells such as human colon cancer SW620 and HCT-8 cells, human melanoma A375 cells and human prostate cancer DU145 and PC-3 cells. Additionally, the tocotrienol-induced paraptosis may occur with autophagy and apoptosis concurrently [84]. Paraptosis-inducing compounds could be developed as chemotherapeutic agents for apoptosis-resistant and multiple-drug resistance cancer [83,85]. Therefore, AnTT, γ-T3 and δ-T3 could be novel adjuvant agents for chondrosarcoma that is primarily resistant to chemotherapy.

The molecular mechanism of tocotrienol-induced paraptosis is largely unclear. Tocotrienol-mediated paraptosis is dependent on protein synthesis, ER stress and/or unfolded protein response (UPR). Tocotrienol-induced paraptosis and cytoplasmic vacuolation can be inhibited by protein synthesis inhibitor and ER stress inhibitor [83,86]. Additionally, tocotrienols including γ-T3 and δ-T3 were previously reported to induce ER stress and UPR in several cancer cells including mouse mammary tumour +SA cells [26], human cervical cancer Hela cells [87], human melanoma BLM and A375 cells [19] and human prostate cancer PC-3 and DU145 cells [84], where it may be related with paraptotic cell death. UPR signalling pathway was triggered as evidenced by the activation of protein kinase-like endoplasmic reticulum kinase (PERK), activating transcription factor 6 and/or inositol-requiring enzyme 1 (IRE1) signalling [88]. γ-T3 was reported to induce UPR with the activation of the PERK and/or IRE1 signalling on mouse mammary tumour +SA cells [26] and human breast cancer MDA-MB-231 and MCF-7 cells [27,89,90]. In addition, the suppression of canonical Wnt signalling pathway may be involved in modulating tocotrienol-induced paraptosis [82]. γ-T3 and δ-T3 were also reported to suppress canonical Wnt signalling in human breast cancer MDA-MB-231 cells [29] and human colon cancer SW620 [30].

The current transcriptome analyses indicated the involvement of ER stress and UPR signalling pathway in tocotrienol-treated SW1353 cells. GO enrichment analysis revealed upregulation of ER and Golgi-related activities, positive regulation of transcription from RNA polymerase II promoter in response to stress, and UPR signalling pathway upon 24 h of AnTT, γ-T3 and δ-T3 treatment. In addition, tocotrienol treatments may be closely related to the occurrence of ER stress, UPR and autophagy processes. Several GO terms such as macroautophagy, protein catabolic process, ER overload response, responses to ER stress, autophagosome assembly and endoplasmic reticulum UPR were significantly upregulated upon 24 h of AnTT, γ-T3 and δ-T3 treatment. This is coherent with the previous studies on the tocotrienol-induced ER stress, UPR and/or autophagy in several cancer cells. On the other hand, the GO terms such as cell division, proliferation and migration were significantly downregulated. This observation is coherent with our findings on the tocotrienol-induced SW1353 growth arrest and apoptosis. In addition, the canonical Wnt signalling pathway was also significantly downregulated upon tocotrienol treatment. This action further supports the speculation on the involvement of Wnt signalling in tocotrienol-mediated paraptosis.

Furthermore, the significantly enriched KEGG pathways were similar among SW1353 cells treated with AnTT, γ-T3 and δ-T3. The enriched pathways related to protein synthesis or posttranslational modifications, such as protein processing in ER, protein export, and SNARE interactions in vesicular transport were upregulated upon tocotrienol treatment. These upregulated pathways may be closely related to the tocotrienol-mediated paraptosis processing. On the other hand, the enriched pathways for carcinogenesis and cell proliferation, such as several pathways in cancer, Hippo signalling pathway, proteoglycan in cancer, cell cycle, Rap1 signalling pathway, viral carcinogenesis, MAPK signalling pathway, PI3k-Akt signalling pathway, Ras signalling pathway and TGF-beta signalling pathway were significantly downregulated. Wnt signalling pathway was also significant downregulated according to KEGG pathway analysis in tocotrienol treatments. Collectively, these pathways are closely related to the anticancer and antiproliferative properties of tocotrienol in chondrosarcoma cells. Additionally, the pathways like focal adhesion, gap junction, adherens junction and regulation of actin cytoskeleton were also significantly downregulated, which may explain the abnormal morphological changes upon tocotrienol treatment.

## 5. Conclusions

AnTT, γ-T3 and δ-T3 decreased the viability of human chondrosarcoma SW1353 cells with an early phase of G1 arrest and apoptosis events at the later stage. δ-T3 is the most potent isomer with anticancer properties, followed by AnTT and γ-T3. The transcriptomic analysis also coherently demonstrated that tocotrienol treatments upregulated signalling pathways in endoplasmic reticulum stress, transcription in response to stress, unfolded protein response, and autophagy. On the other hand, cell proliferation and cancer-related pathways, such as pathway in cancer, Hippo signalling pathway and Wnt signalling pathway, were significantly downregulated upon tocotrienol treatment. Interestingly, tocotrienol treatment may induce paraptosis or paraptosis-like cell death as evidenced by the massive cytoplasmic vacuole formation on the treated SW1353 cells. Paraptotic cell death is clinically important in the treatment of apoptosis-resistant cancer like chondrosarcoma. The anticancer effects of AnTT, γ-T3 and δ-T3 render them suitable candidates in chondrosarcoma adjuvant therapy, pending validation from more intensive research.

## 6. Patents

A patient application has been filed for the use of tocotrienol adjuvant therapy for chondrosarcoma (file number: UKM.IKB.800-4/1/3551).

## Figures and Tables

**Figure 1 nutrients-14-04277-f001:**
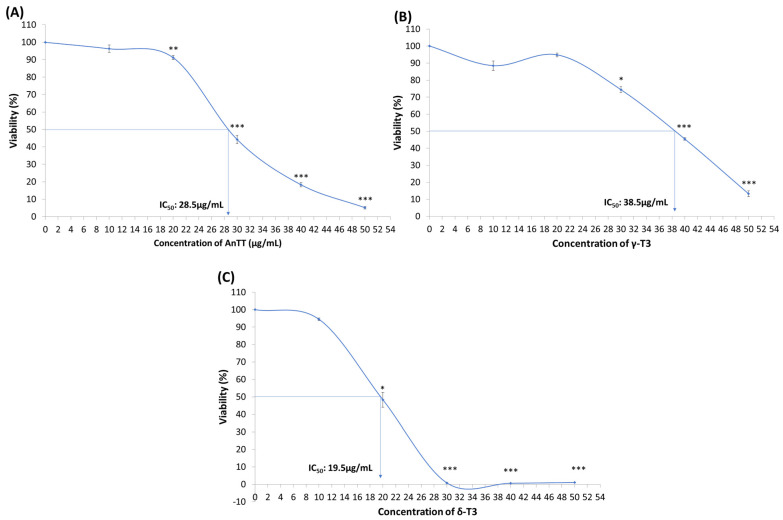
The cytotoxicity of AnTT (**A**), γ-T3 (**B**) and δ-T3 (**C**) on SW1353 cells upon 24 h treatment. *, ** and *** indicate a significant difference of *p* < 0.05, *p* < 0.01 and *p* < 0.001 compared to VC. The viability of VC is assumed as 100%. Abbreviations: AnTT, annatto tocotrienol; δ-T3, δ-tocotrienol; γ-T3, γ-tocotrienol; IC_50_, half-maximal inhibitory concentration; VC, vehicle control.

**Figure 2 nutrients-14-04277-f002:**
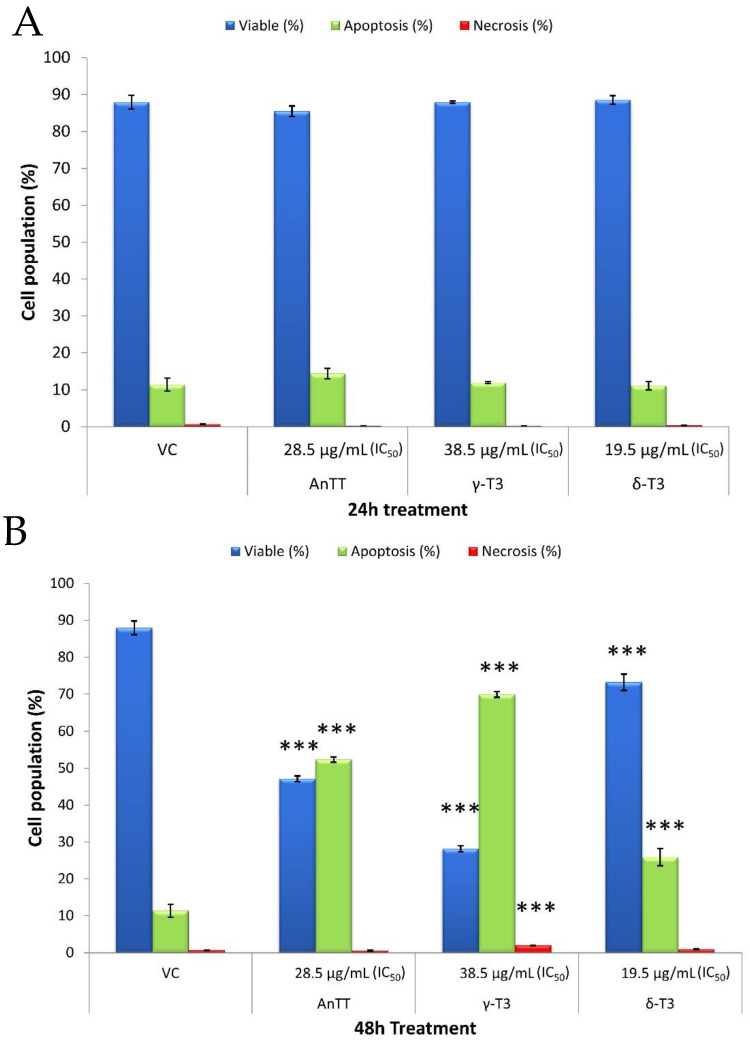
Mode of cell death in SW1353 cells over a 24 h (**A**) and 48 h (**B**) treatment with AnTT, γ-T3 and δ-T3 at their respective IC_50_. Although 24 h treatment of tocotrienols significantly reduces the viability of cells, they do not induce cell death. Tocotrienols-induced growth arrest during 24 h treatment is postulated. However, prolonging the treatment time to 48 h did induce significant cell death. *** indicate a significant difference of *p* < 0.001 compared to VC. Abbreviations: AnTT, annatto tocotrienol; δ-T3, δ-tocotrienol; γ-T3, γ-tocotrienol; IC_50_, half-maximal inhibitory concentration; VC, vehicle control.

**Figure 3 nutrients-14-04277-f003:**
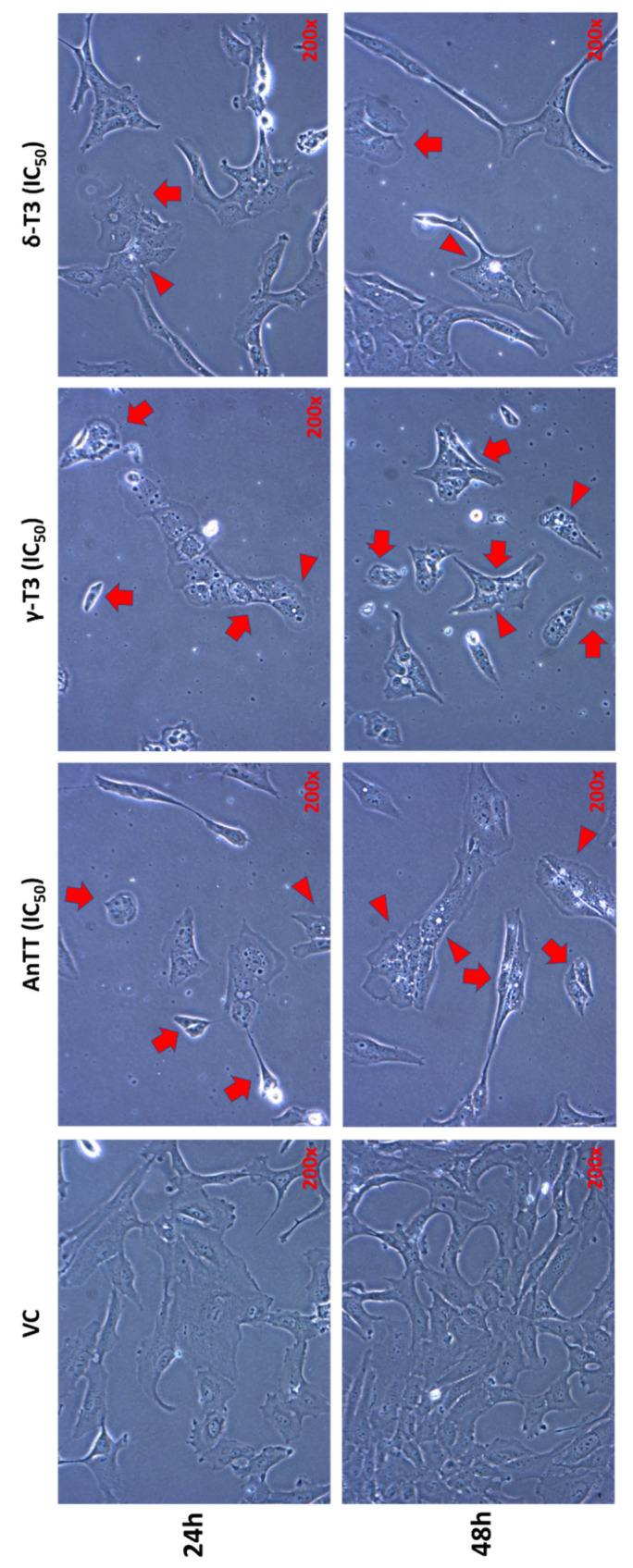
Morphological changes of SW1353 cells upon 24 h and 48 h AnTT, γ-T3 and δ-T3 treatment. SW1353 cells in the VC group exhibit typical spindle-shaped adherent cell morphology. Tocotrienol-treated SW1353 cells show abnormal morphological changes (arrow), such as cell shrinkage, elongated or squamous cell shape, and massive cytoplasmic vacuolation (arrowhead), especially upon 48 h treatment. Abbreviations: AnTT, annatto tocotrienol; δ-T3, δ-tocotrienol; γ-T3, γ-tocotrienol; IC_50_, half-maximal inhibitory concentration; VC, vehicle control.

**Figure 4 nutrients-14-04277-f004:**
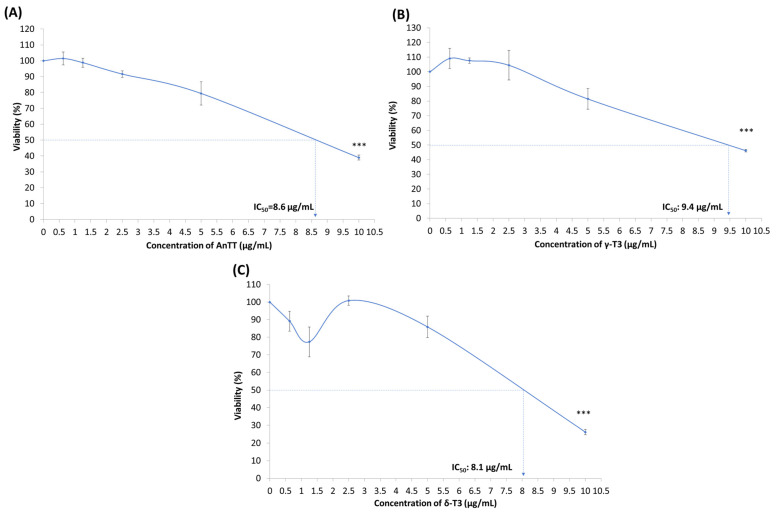
The antiproliferative activities of AnTT (**A**), γ-T3 (**B**) and δ-T3 (**C**) on SW1353 cells upon 72 h treatment. *** indicates a significant difference of *p* < 0.001 compared to VC. The viability of VC is assumed as 100%. Abbreviations: AnTT, annatto tocotrienol; δ-T3, δ-tocotrienol; γ-T3, γ-tocotrienol; IC_50_, half-maximal inhibitory concentration; VC, vehicle control.

**Figure 5 nutrients-14-04277-f005:**
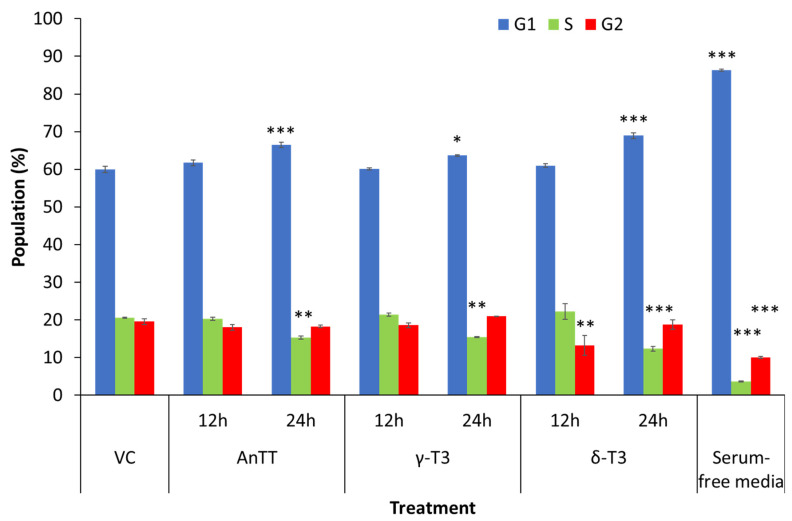
The quantitative analysis of cell cycle distribution after tocotrienol treatment for 12 and 24 h. AnTT, γ-T3 and δ-T3 in their IC_50_ concentration induced G1 arrest on SW1353 chondrocytes upon 24 h treatment with the lower S or G2-phase percentage. The shorter treatment time of 12 h did not significantly alter the cell cycle distribution. A 24 h serum-free condition was performed to arrest the SW1353 cells at the G1 phase. *, ** and *** indicate significant differences of *p* < 0.05, *p* < 0.01 and *p* < 0.001 compared to VC. Abbreviations: AnTT, annatto tocotrienol; δ-T3, δ-tocotrienol; γ-T3, γ-tocotrienol; VC, vehicle control.

**Figure 6 nutrients-14-04277-f006:**
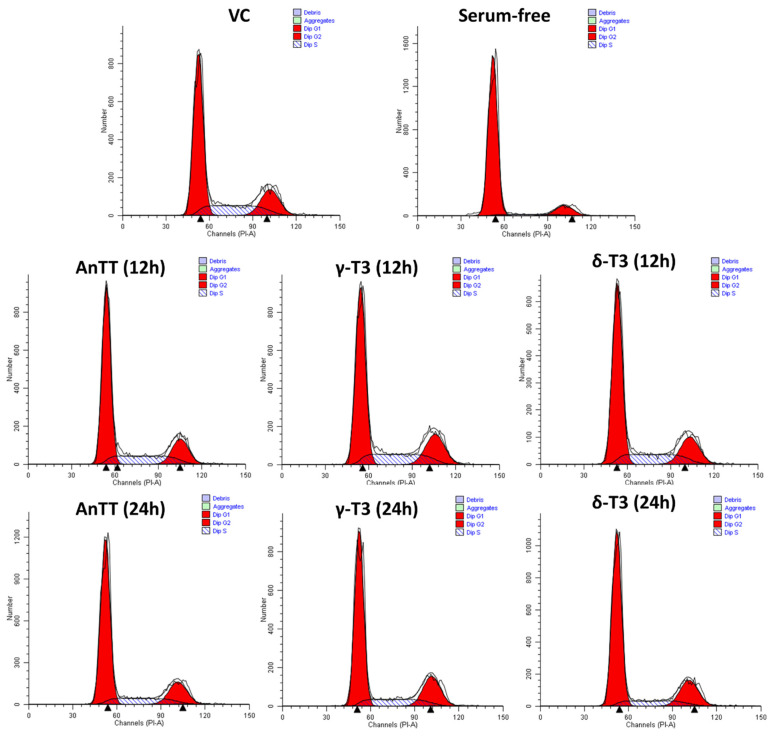
The representative cell cycle distribution analysis from the ModFit LT software. Abbreviations: AnTT, annatto tocotrienol; δ-T3, δ-tocotrienol; γ-T3, γ-tocotrienol; VC, vehicle control.

**Figure 7 nutrients-14-04277-f007:**
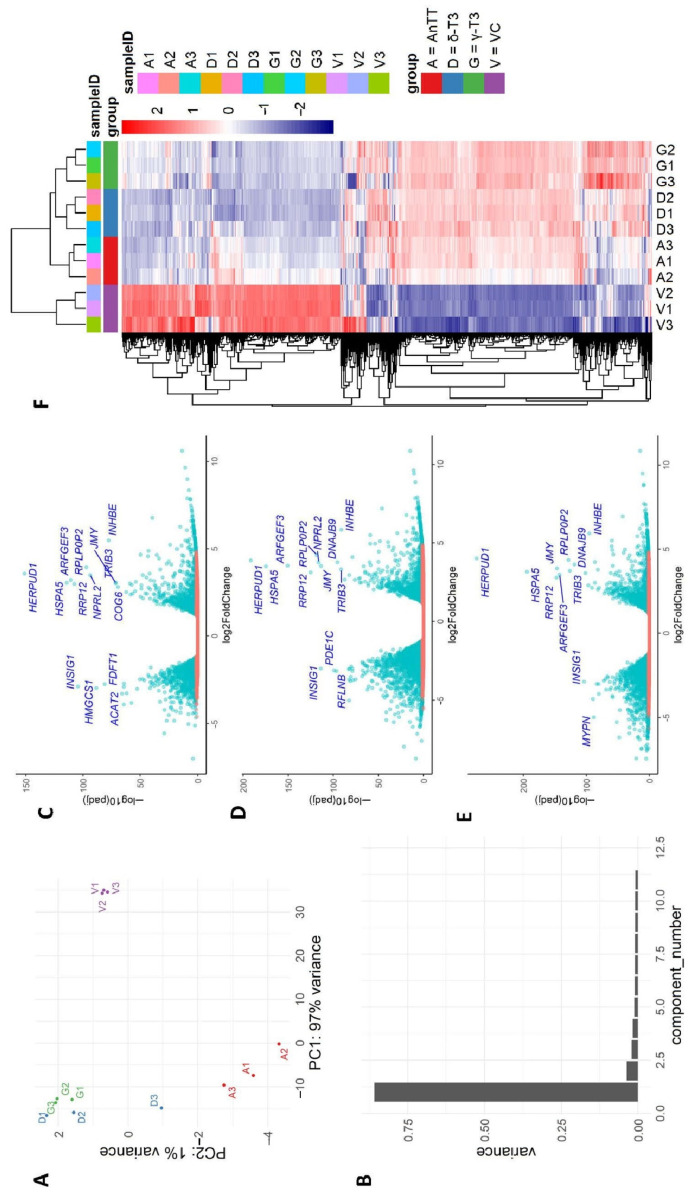
Tocotrienol treatments showed similar gene expression variation when compared to the control group. (**A**) Principal component analysis (PCA) plot depicting the two largest components of variance in gene expression in the three treatment groups. (**B**) Scree plot showing the proportion of variance explained by the first 12 principal components (PC1 explains 97% of the total variance). (**C**–**E**) Volcano plots visualizing DEGs between pairwise comparisons of different tocotrienol treatments and vehicle control. Green points represent significant DEGs with an adjusted *p*-value < 0.05. Top DEGs were selectively labelled in blue text. (**F**) Hierarchical clustering and heatmap of the top 1000 differential expressed genes, showing clear discrepancies in expression profiles between the tocotrienol treatments (annatto tocotrienol (AnTT), δ-tocotrienol (δ-T3), and γ-tocotrienol (γ-T3)) and the vehicle control (VC) group.

**Figure 8 nutrients-14-04277-f008:**
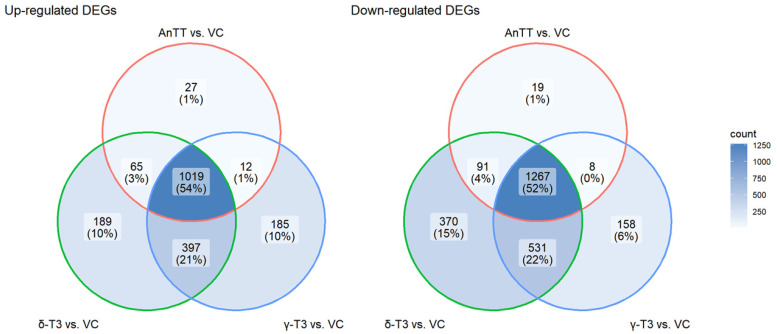
Venn Diagrams show the numbers of the up- and down-regulated differential expressed genes (DEGs) among the pairwise comparisons of different tocotrienol treatments (namely annatto tocotrienol (AnTT), δ-tocotrienol (δ-T3) and γ-tocotrienol (γ-T3)) and vehicle control (VC) group.

**Figure 9 nutrients-14-04277-f009:**
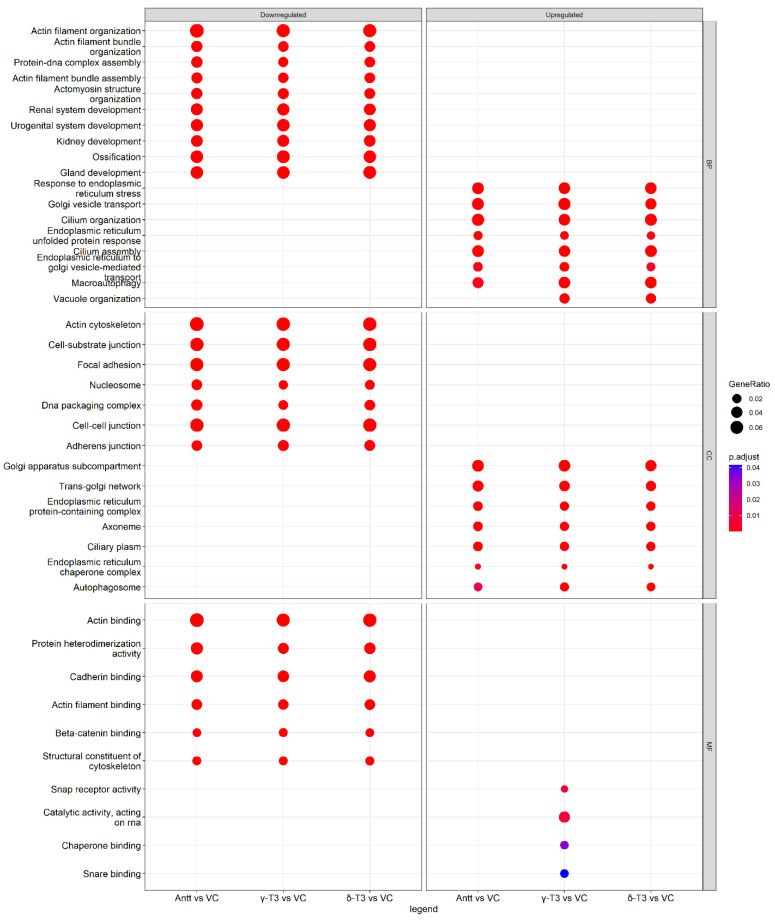
Gene Ontology enrichment of tocotrienol treatments against vehicle control. Pathway enrichment analysis of AnTT versus VC, δ-T3 versus VC, and γ-T3 versus VC, were performed using gene set over-representation analysis using clusterProfiler R package. Abbreviations: BP, biological process; CC, cellular components; MF, molecular function; AnTT, annatto tocotrienol; δ-T3, δ-tocotrienol; γ-T3, γ-tocotrienol; VC, vehicle control.

**Figure 10 nutrients-14-04277-f010:**
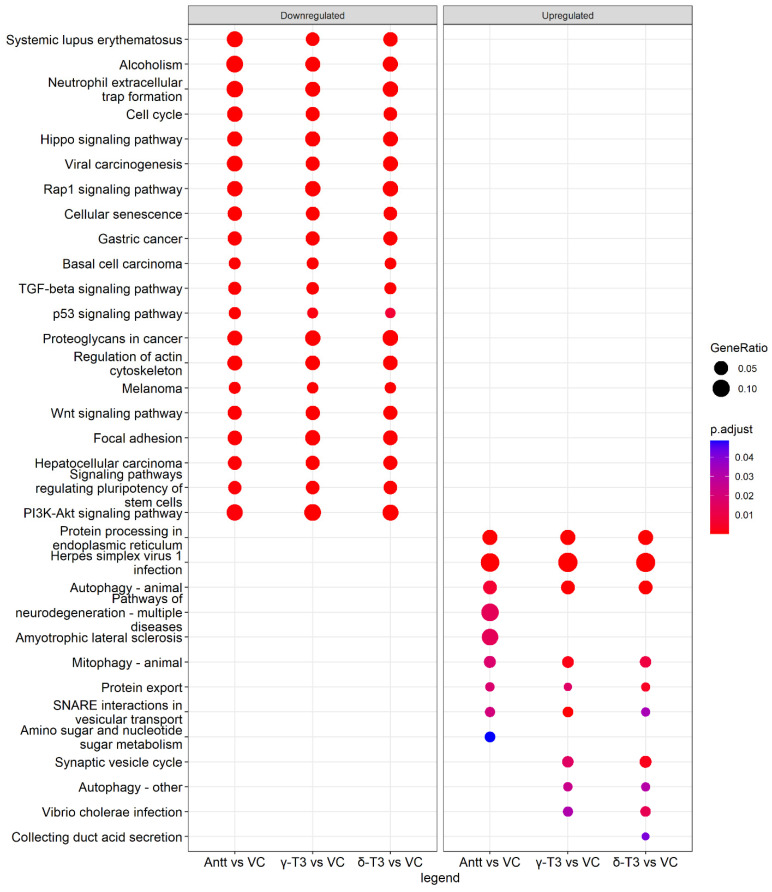
The KEGG enriched pathways with the upregulated DEGs for AnTT, γ-T3 and δ-T3 treatment as compared to VC. Abbreviations: Antt, annatto tocotrienol; δ-T3, δ-tocotrienol; γ-T3, γ-tocotrienol; DEGs, differentially expressed genes; VC, vehicle control.

**Figure 11 nutrients-14-04277-f011:**
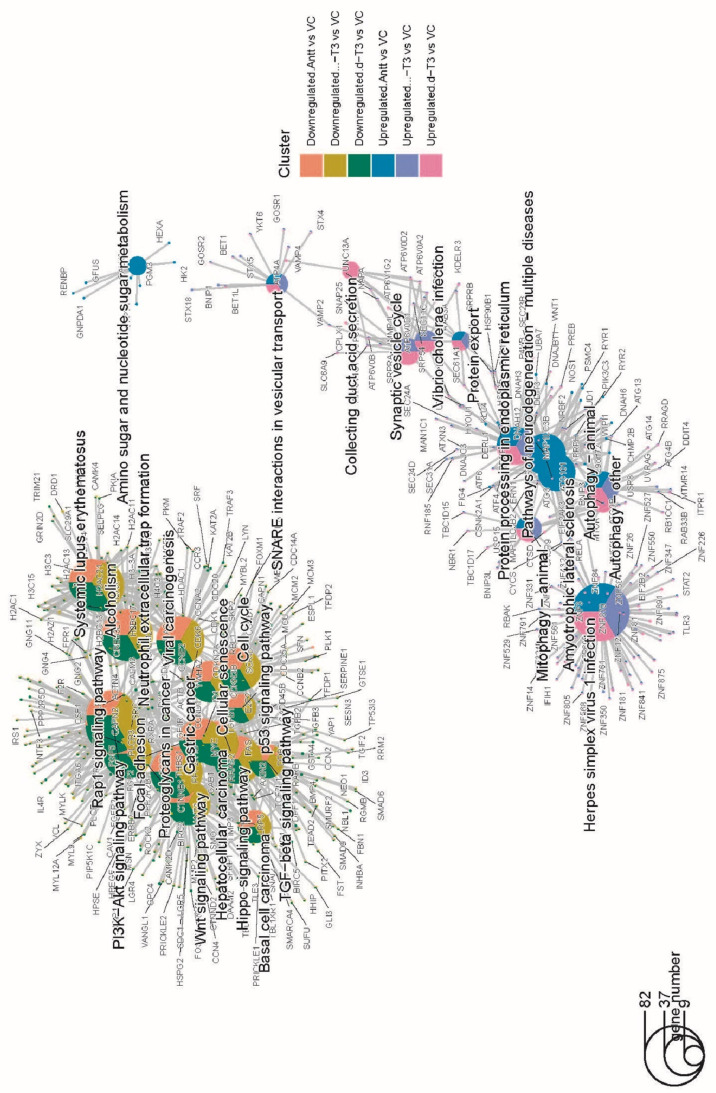
Cnetplot of KEGG pathways showing up- and down-regulated DEGs of tocotrienol treatments enriched in different pathways. The symbol adjacent to nodes represents the specific gene. The colour bar represents the up- or down-regulated DEG datasets for each treatment group.

**Figure 12 nutrients-14-04277-f012:**
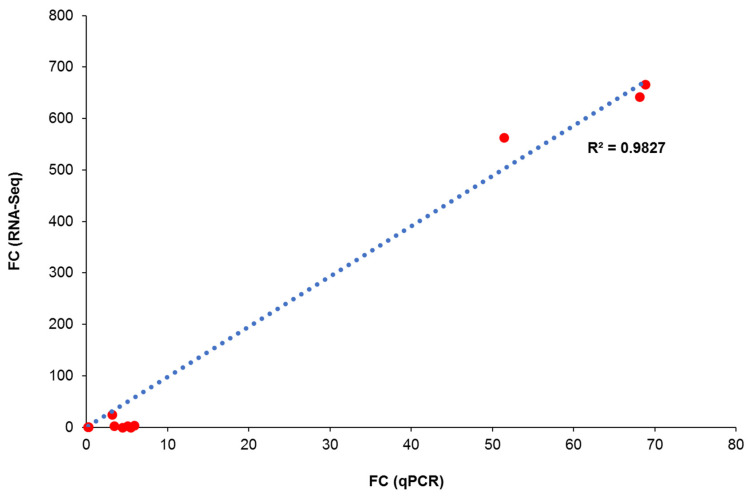
Correlation of FC values from qPCR and RNA sequencing for 4 target genes. Each dot represents a pair of data with FC values from respective qPCR and RNA sequencing. Abbreviations: FC, fold change, R^2^, correlation coefficient.

**Table 1 nutrients-14-04277-t001:** Comparison of qPCR and RNA sequencing data in FC values for 4 target genes.

Gene ID	AnTT vs. VC		δ-T3 vs. VC		γ-T3 vs. VC		AnTT vs. δ-T3		AnTT vs. γ-T3		δ-T3 vs. γ-T3	
	FC (RNA-seq)	FC (qPCR)	FC (RNA-seq)	FC (qPCR)	FC (RNA-seq)	FC (qPCR)	FC (RNA-seq)	FC (qPCR)	FC (RNA-seq)	FC (qPCR)	FC (RNA-seq)	FC (qPCR)
ARMCX3	2.42	3.42	2.98	5.02	3.55	5.92	N.A.	0.68	N.A.	0.58	N.A.	0.85
HMGCS1	0.13	0.02	0.14	0.21	0.15	0.23	N.A.	0.94	N.A.	0.85	N.A.	0.91
ADAMTSL1	0.04	3.06	0.03	4.33	0.01	5.20	N.A.	0.70	N.A.	0.59	N.A.	0.83
KLHDC7B	562.64	45.92	666.12	66.26	642.02	62.92	N.A.	0.69	N.A.	0.73	N.A.	1.05

Abbreviations: AnTT, annatto tocotrienol, δ-T3, δ-tocotrienol, γ-T3, γ-tocotrienol, FC, fold change, N.A., not available.

**Table 2 nutrients-14-04277-t002:** IC_50_ values of AnTT, γ-T3 and δ-T3 from previous studies.

Tocotrienols	Cells	IC_50_ Values (μg/mL)	References
AnTT	Human breast cancer SKBR3 cells	24 h: 14.33 *	[65]
	Human prostate cancer LNCaP cells	24 h: <1.99 *	[66]
	Human prostate cancer PC-3 cells	48 h: <10	[67]
δ-T3	Human breast cancer SKBR3 cells	24 h: 8.73	[65]
	Human breast cancer MDA-MB-231 cells	72 h: 6.9	[68]
	Human breast cancer MCF-7 cells	72 h: 6.8	
	Human prostate cancer LNCap cells	24 h: 22.61	[69]
		48 h: 21.02	
		72 h: 13.48	
	Human prostate cancer PC-3 cells	24 h: 23.40	
		48 h: 7.14	
		72 h: 3.57	
	Human lung cancer A549 cells	24 h: 0.68	[70]
		48 h: 0.63	
		72 h: 0.50	
	Human glioma U87MG cells	24 h: 0.58	
		48 h: 0.51	
		72 h: 0.45	
	Human liver cancer HepG2 cells	72 h: 3.81	[71]
	Human liver cancer HCT116 cells	48 h: 7.93	[72]
	Human liver cancer HT29 cells	48 h: 11.90	
	Human melanoma A2058 cells	72 h: 14.87	[73]
	Human melanoma A375 cells	72 h: 8.84	
	Human pancreatic cancer MiaPaCa-2 cells	72 h: 19.83	[74]
γ-T3	Human breast cancer MDA-MB-231	24 h: 16.01 48 h: 12.73	[75]
		72 h: 4.7	[68]
	Human breast cancer MCF-7 cells	24 h: 16.88 48 h: 13.51	[75]
		72 h: 6.35	[68]
	Human prostate cancer LNCap cells	24 h: 24.23	[69]
		48 h: 20.12	
		72 h: 18.48	
	Human prostate cancer PC-3 cells	24 h: 21.76	
		48 h: 4.11	
		72 h: 3.28	
	Human lung cancer A549 cells	24 h: 1.17	[70]
		48 h: 1.05	
		72 h: 0.71	
	Human glioma U87MG cells	24 h: 2.06	
		48 h: 1.54	
		72 h: 1.36	
	Human liver cancer HepG2 cells	72 h: 10.14	[71]
	Human liver cancer HCT116 cells	48 h: 7.19	[72]
	Human liver cancer HT29 cells	48 h: 12.32	

* Original studies had reported the IC_50_ values in μM. The average molecular weight of tocotrienols (398 g/mol) was used in converting to μg/mL based on the mixture of 90% of δ-T3 (396.6 g/mol) and 10% of γ-T3 (410.6 g/mol).

## Data Availability

The data of this study are available at reasonable request from the corresponding author.

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
