# Peer review of "Transcriptomic Analysis of the Anticancer Effects of Annatto Tocotrienol, Delta-Tocotrienol and Gamma-Tocotrienol on Chondrosarcoma Cells"

_nutrients, 2022, doi:10.3390/nu14204277_

Round 1

Reviewer 1 Report

In the article entitled: Transcriptomic analysis of the anticancer effects of annatto tocotrienol, delta-tocotrienol, and gamma-tocotrienol on chondrosarcoma cells authors tried to prove the anticancer ability of these natural compounds. First of all, the cell target for the presented studies has been very well focused on chondrosarcoma is a good example of apoptosis-resistant cancer. Secondly, tocochromanols are commonly present in our nutrition, also as a technological additive – antioxidant Vit E. However, reduction of fat consumption reduces the vitamin ADEK fortification. All the above can leads to significant physiological negative outcomes. The article is well-written and readable. However, figures 11 and 7 should be in high resolution, in current forms that are unclear. The methodology and methods have been presented correctly. The conclusion should be extended with medical importance not only for scientists but also for common people. It would be a wonder if authors put the information about frequencies of chondrosarcoma.

Author Response

Dear reviewer, 

We appreciate the constructive comments and have responded to each of them in the attached response sheet.

Reviewer 2 Report

The manuscript “Transcriptomic analysis of the anticancer effects of annatto tocotrienol, delta-tocotrienol, and gamma-tocotrienol on chondrosarcoma cells”. The study has good objectives, however, several points need to address.

1.       In the abstract section name, some on the important upregulated markers for apoptosis and autophagy observed in the present work in response to the treatment of tocochromonal

2.       Delete the word “reviewed in” (line 65)

3.       Rewrite the sentence “Briefly……………. tocotrienol for another 24h” (line 152-155), Also mention the exact range of the used concentration.

4.       Write 570 nM as 570 nm (line 157).

5.       Mention how the IC50 was calculated is it by linear equation or by the logarithmic equation (line 160)

6.       Write 200 g as 200xg (line 170, 197 and 213, Similarly change 400 g (line 200) and 8000 g (line 216, 218, 220), and 1000 g (222, 24, 227) as 400x g, 8000xg and 10000xg. Also, check this through in the entire MS.

7.       What does the author mean by the modified MTT assay???? Better merge this section with MTT assay, Also, mention accurately what series of concentrations were used.

8.       Line197-198 is not clear, elaborate the process for clarity.

9.       Make a proper statement for “5 x 104 cells/mL; 6.5 mL of cells” (line 209)

10.   Mention the name of the 5 selected genes with 4 genes random (line 267-268)

11.   In Figure 1 statistical results are compared with the VC, therefore adding the value obtained from the VC-treated cells (fig 1. A, B, C). Also, change the X-axis concentration in the range of 10-50 rather than 2-54.

12.   What is the unit of concentration In figure 1 Band C? Add the unit in the respective place.

13.   In figure 2 remove the word IC50 after the used concentrations.

14.   In figure 3 add the scale bar.

15.   What is the need for section 3.44 better merge I with the MTT section?

16.   Why the IC 50 value of the same compounds against the same cell line is different? Usually, the Ic 50 is the characteristic value of a particular compound against the particular cells in the defined set of conditions. Please justify this.

17.   In figure 4 add the value of VC and then compared the statistical results.

18.   Add the % value of PC1 and PC2 in the figure…

19.   Improve the resolution of figure 11, its hard to read.

Author Response

Dear reviewer,

We appreciate the constructive comments and have responded to each of them in the attached response sheet. Thank you.

Round 2

Reviewer 2 Report

Asked changes have been added in the revised manuscript.